# More than Just a Toothache: Inflammatory Mechanisms Linking Periodontal Disease to Cardiovascular Disease and the Protective Impact of Cardiorespiratory Fitness

**DOI:** 10.3390/biomedicines13071512

**Published:** 2025-06-20

**Authors:** Michael F. Mendoza, Michael A. Anzelmo, Nina M. Suan, Chloe S. Cuccia, Carl J. Lavie

**Affiliations:** 1The Gayle and Tom Benson Cancer Center, Ochsner Clinic Foundation, New Orleans, LA 70121, USA; 2Department of Internal Medicine, Ochsner Clinic Foundation, New Orleans, LA 70124, USA; 3Faculty of Medicine and Surgery, University of Santo Tomas, Manila 1015, Philippines; nnsuan@gmail.com; 4Speech-Language Pathology and Audiology Department, Southern University, Baton Rouge, LA 70813, USA; chloescuccia@gmail.com; 5Department of Cardiology, Ochsner Clinic Foundation, New Orleans, LA 70124, USA; 6Ochsner Clinical School, The University of Queensland Medical School, New Orleans, LA 70121, USA

**Keywords:** cardiovascular disease, periodontal disease, periodontitis, systemic inflammation, oral health, mortality, cardiorespiratory fitness

## Abstract

Periodontal disease is a chronic inflammatory condition that extends beyond the oral cavity and is increasingly recognized as a significant contributor to cardiovascular disease, the leading cause of global mortality. Systemic inflammation serves as the key link between the two, with elevated biomarkers such as C-reactive protein and interleukin-6 observed in individuals with PD, driving endothelial dysfunction and atherogenesis. Epidemiological and mechanistic studies consistently demonstrate that PD not only heightens the risk of major adverse cardiovascular events but is also independently associated with increased all-cause and cardiovascular mortality. In parallel, emerging research underscores the protective role of regular physical activity and improved cardiorespiratory fitness, which attenuate systemic inflammation and may reduce the burden of both PD and CVD. This review integrates current evidence on the shared inflammatory mechanisms linking PD and CVD, highlights the systemic impact of poor oral health, and advocates for incorporating exercise-based interventions and oral health assessments into comprehensive cardiovascular risk management. A multidisciplinary approach—bridging cardiology, dentistry, and preventive medicine—may offer a novel strategy to reduce inflammation-driven morbidity and mortality.

## 1. Introduction

“Can your dentist save your heart?” Periodontal disease or periodontitis (PD), a chronic inflammatory condition affecting the gums and supporting structures of teeth, has been increasingly recognized as a significant contributor to the progression of cardiovascular disease (CVD)—the leading cause of death worldwide. This growing awareness has prompted healthcare providers to address multiple modifiable risk factors to mitigate CVD risk [1]. Among these, systemic inflammation (SI) has emerged as a central and actionable driver. SI is not only a marker of disease activity (e.g., C-reactive protein [CRP]) but also a causative factor that accelerates lesion formation and progression in coronary artery disease (CAD), affecting both stable and unstable atheroma. The pathogenic cascade begins when small lipid particles (<70 nm) penetrate the endothelium and form atheroma, which, under persistent inflammatory conditions, become unstable and potentially obstruct distal vessels [2,3,4,5].

Common CVD risk factors such as high blood pressure, elevated cholesterol, diabetes (DM), and smoking all contribute to elevated SI and, consequently, to atherosclerosis progression and increased adverse cardiovascular (CV) events [6]. Studies have shown that PD can also increase the risk of CVD development through the same shared inflammatory risk factor [7,8,9]. Similarly, PD has been implicated in promoting CVD via this shared inflammatory pathway. In PD, chronic infection due to poor oral hygiene (OH) induces local gum inflammation that progresses to SI, contributing to endothelial dysfunction and atherosclerosis [5,6,7,9,10]. A systematic review and meta-analysis conducted in 2021 reinforced this inflammatory link, revealing that individuals with chronic PD had an average increase in CRP levels of 103% compared to controls, while those with aggressive PD demonstrated up to a 191% increase [11].

The public health implications of these findings are substantial, given that PD affects approximately 796 million people globally and roughly 42% of U.S. adults aged 30 and above, including 7.8% with severe disease [1]. Despite its prevalence, PD remains underdiagnosed and undertreated—similar to hypertension (HTN), a well-established modifiable CVD risk factor [12]. PD’s role in SI underscores its contribution to subclinical vascular injury and major adverse CV outcomes, positioning it as a passive yet potent cardiovascular risk factor. Consequently, oral health status should be viewed as a vital clinical indicator of systemic inflammatory burden and CV risk. Improved oral hygiene and targeted periodontal interventions offer a promising strategy to reduce SI and, in turn, lower the risk of CVD progression. Clinicians, particularly those managing patients at risk for acute CV events, should consider incorporating routine assessments of OH into comprehensive prevention plans [13]. This review will therefore explore the following: (1) the relationship between PD, CVD, and mortality; (2) the mechanistic role of SI in CVD progression; (3) the impact of fitness on inflammation, PD, and CV outcomes; and (4) future clinical and public health applications.

## 2. Periodontitis and CVD: Evidence from Population Studies, Imaging and Animal Models

Given the strong evidence linking PD to both SI and CVD progression, it is critical to recognize its role as a modifiable driver of cardiovascular mortality. In the U.S. alone, cardiovascular disease accounts for one in every five deaths—over 931,000 lives in 2021 alone—with one person dying every 33 s from CVD [14,15].

Large cohort studies have identified a significant link between PD and an increased risk of developing CVD, with observational studies providing significant evidence of this link. A 2021 meta-analysis published in Frontiers in Cardiovascular Medicine examined 26 studies to assess the association between PD and CVD. The findings revealed that individuals with PD had a higher prevalence of CVD, with 7.2% of PD patients affected. That study also found a significant association between PD and CVD in both men (odds ratio [OR] = 1.22; 95% confidence interval [CI]: 1.12–1.34) and women (OR = 1.11; 95% CI: 1.05–1.17), indicating that PD is linked to an increased risk of CVD regardless of sex. The authors also concluded that interventions targeting PD may be beneficial for CV health [16].

From 2019 to 2023, approximately five randomized and non-randomized clinical studies were published evaluating the potential connection between PD and CVD. While none of these studies established a definitive causal relationship, some demonstrated that treatment of PD can lead to improvements in surrogate markers of CVD, such as reduced SI and improved endothelial function [17]. A 2012 review by Lockhart et al. [18] analyzed 57 observational studies examining the association between PD and CVD. Their analysis revealed that individuals with PD had a 1.24-fold increased risk of developing CVD, even after adjusting for confounding factors such as smoking and DM. Notably, their study found that ischemic stroke had the strongest association with PD, with an OR of 2.85 (95% CI: 1.56 to 5.20, *p* < 0.001).

Between 1998 and 2022, systematic reviews and meta-analyses evaluated the relationship between PD and CVD. A study by Vedin et al. [19] examined the association between PD and CVD in patients with stable coronary heart disease (CHD). They reported that individuals with PD had a relative risk (RR) of 1.14 (95% CI: 1.07 to 1.21, *p* < 0.001) for developing CVD compared to those without PD. In those with severe PD, the risk further increased to RR = 1.34 (95% CI: 1.21 to 1.48, *p* < 0.001). A consensus report by Sanz et al. confirmed a significant association between PD and CVD, further elucidating that the findings similarly suggest a dose-response relationship, where the severity of PD correlates with an increased risk of CVD [20]. A pooled meta-analysis in 2023, which included 39 cohort studies (*n* = 4.3 million), found that PD was linked with more major adverse CV events (pooled RR = 1.24). Cardiac death had an RR of 1.42, whereas all-cause mortality, stroke, CHD, myocardial infarction (MI), and cardiac death had an RR equal to 1.31, 1.26, 1.20, and 1.14, respectively [21]. Another meta-analysis published in the 2017 *BMC Cardiovascular Disorders* analyzed 22 observational studies (*n* = 129,630 participants) and found that individuals with PD have an increased risk of MI, with an OR = 2.02 (95% CI: 1.59 to 2.57) [22].

Evidence was derived from non-invasive imaging techniques and invasive animal studies. One established imaging marker used to assess early atherosclerosis is carotid intima–media thickness (cIMT) [5,23]. Several studies have demonstrated that individuals with PD independently exhibited increased cIMT measurements, which evidenced a direct correlation between subclinical atherosclerosis and the severity of PD. The CHIEF study found that individuals with moderate to severe PD had consistently elevated cIMT values compared to healthy controls, even after adjusting for traditional CV risk factors like smoking, HTN, and DM. This association persisted across populations and analyses [24,25,26].

FDG-PET/CT imaging further reinforces this association, where increased FDG uptake in periodontal regions has been linked to increased macrophage infiltration in carotid/coronary artery plaques, which was suggestive of a systemic inflammatory response of oral origin (*p* < 0.001). Additional evidence from pharmacologic studies shows that high-dose atorvastatin reduces both arterial inflammation and PD severity, reinforcing the shared inflammatory pathway [27,28,29,30]. In animal studies, inoculating animals with *Porphyromonas gingivalis* induces atherosclerosis via plaque formation, lipid accumulation, and aortic lesions [31,32,33]. Moreover, ligature of the teeth in rat models induced periodontal disease, which later showed increased oxidative stress and lipid deposits in the aorta, reinforcing the role of PD in accelerating atherogenesis [34]. While animal models offer compelling mechanistic insights, applying them to human populations remains complex due to confounding variables. Nevertheless, these findings emphasize the global significance of PD as something more than a dental issue but also as a key contributor to SI, CVD progression, and mortality.

## 3. Periodontitis and Mortality

Comorbidities such as type 2 diabetes mellitus (T2DM) significantly amplify the risk of mortality. A study by Ngamdu et al. (2022) [35], analyzing National Health and Nutrition Examination Survey (NHANES) 2013–2014 data (*n* = 2830 adults aged ≥30 years), found that individuals with both T2DM and PD had a 1.66-fold higher risk of CV-related mortality compared to those with T2DM but no history of PD (*p* = 0.03). Their study further revealed that those with severe PD had more than triple the odds of developing CVD compared to individuals with only mild PD (OR = 3.59; 95% CI: 1.12–11.54, *p* = 0.03). Complementary to this, research from Chen et al. (2024) indicates that individuals with more severe forms of PD experience even higher mortality rates, wherein the group found that individuals with CVD and moderate to severe PD had a 25% higher all-cause mortality rate compared to those with mild or no PD (HR: 1.25; 95% CI: 1.02–1.52; *p* = 0.03) [36].

Of note, smoking is a well-established risk factor that contributes to both periodontal disease severity and cardiovascular morbidity. It promotes oral microbial dysbiosis, impairs the immune response, and reduces gingival blood flow, which can worsen periodontal attachment loss and pocket depth. Simultaneously, smoking accelerates atherosclerosis via endothelial injury, oxidative stress, and increased systemic inflammation. A meta-analysis by Nazir et al. (2020) reported that smokers were nearly twice as likely to develop periodontitis (OR = 1.82; 95% CI: 1.63–2.02), and the severity of PD was positively associated with cumulative smoking exposure (pack-years) [37].

Furthermore, the INTERHEART study showed that current smokers had a threefold increased risk of myocardial infarction, with risk magnitude increasing linearly with smoking intensity (*p* < 0.001) [38].

Additionally, a meta-analysis reported that individuals with severe PD had a 40% higher risk of mortality, regardless of the underlying cause of death [39]. Concordantly, previous studies from 2020 indicated that severe PD adds one point to the side and a multiple-times higher risk of CVD-related mortality compared to those with mild or no evidence of PD (HR: 1.7; 95% CI: 1.2–2.4; *p* < 0.01) [40]. A recent retrospective cohort study by Larvin et al. (2024) investigated the association between PD and both all-cause and cause-specific mortality using adult participants recruited from six National Health and Nutrition Examination Survey cycles (1999–2014) [41]. Their study included 15,030 participants who were followed up for nine years and revealed a significant increase in all-cause mortality among those with PD compared to the control group (HR: 1.22; 95% CI: 1.12–1.31; *p* < 0.01) [41]. To consolidate these pooled data, a meta-analysis of 39 studies by Guo et al. (2023) also showed that individuals with severe PD had a 40% higher risk of all-cause mortality RR = 1.40 (95% CI: 1.15–1.69), indicating a dose-response relationship when it comes to the degree of PD [21].

## 4. Mechanism of CVD Progression in Periodontal Disease

The systemic impact of PD provides a compelling backdrop for understanding its role in exacerbating CVD outcomes. PD, a chronic inflammatory condition driven by bacterial biofilms in the oral cavity, triggers a cascade of inflammatory responses, which extends beyond the oral cavity, entering the bloodstream and contributing to endothelial dysfunction, atherosclerosis, and other vascular pathologies.

The oral microbiome can be categorically divided into distinct regions based on both physical and chemical properties of the microenvironment, which influence the growth of local microbiota [39]. The subgingival space is of primary interest due to its relatively thin epithelium, proximity to highly vascular networks, and bacterial accumulation in this space [40,41]. Unlike other commensal environments in the body, where the defense system creates an inhospitable barrier to bacterial translocation—such as the impermeable and keratinized layers of the skin—the oral cavity allows bacteria to enter the circulation.

This can result in transient bacteremia during routine activities like toothbrushing, flossing, and chewing [20]. While these episodes have been associated with SI, most cases of bacteremia are brief and resolved without clinical consequences [20,42]. Of note, Gram-negative species (like *Porphyromonas gingivalis*, *Treponema denticola*, and *Tannerella forsythia*) are the usual species implicated in PD and are of particular interest due to their unique bacterial membrane containing lipopolysaccharides, which can bind to toll-like receptors (TLRs) on oral epithelial cells, triggering a downstream cascade of pro-inflammatory cytokines, including IL-6 [39,43,44]. The elevated production of IL-6 in periodontal tissues leads to its entry into the systemic circulation.

Of note, chronic inflammation plays a crucial role in the development and progression of atherosclerosis, a leading cause of CVD [45,46]. SI not only serves as a marker for advanced CVD but also actively contributes to its progression. These are reflected by elevated levels of inflammatory markers, such as CRP and IL-6, which have been implicated in the increased risk of CHD and CV-related events. Therefore, these acute reactants are surrogate indicators of ongoing SI and drive the progression of CVD. In detail, the key pro-inflammatory cytokine, IL-6, triggers the liver to produce CRP, which, in turn, promotes leukocyte recruitment and inflammatory signaling in endothelial cells, leading to endothelial dysfunction (ED). Elevated IL-6 can induce endothelial and immune cells to upregulate adhesion molecules (e.g., VCAM-1, ICAM-1), allowing for monocyte adhesion and arterial wall entry. This results in macrophage stimulation, allowing for increased LDL uptake in order to form foam cells, which leads to increased production of pro-thrombotic factors that can later form an unstable plaque [47,48,49]. CRP itself can further perpetuate endothelial dysfunction and coagulation. SI shifts the balance in favor of proteinases such as matrix metalloproteinases and certain cathepsins, which in turn contribute to the aforesaid plaque instability and fibrous cap rupture. In recent years, this has been distinguished from plaque erosion, which occurs from shedding that does not involve macrophage-driven cap rupture [50,51,52]. In sum, IL-6 and CRP directly contribute to endothelial injury, plaque development, and plaque rupture [47]. Besides causing vascular inflammation, ED also promotes vasoconstriction by reducing the expression of nitric oxide, prostacyclin, and endothelium-derived hyperpolarizing factor [53,54]. In chronic PD driven by pathogenic oral microorganisms, ED can also result from immune system cross-reactivity. This occurs when the immune system mistakenly identifies its own proteins as foreign bacterial proteins due to structural similarities between bacterial heat-shock proteins (HSPs) and human HSPs. Consequently, the immune response targets both the bacterial proteins and the body’s own endothelial cells, leading to inflammation and vascular damage [54,55]. Therefore, elucidating these molecular pathways can offer mechanistic insight into potential therapeutic targets for mitigating inflammation-driven vascular disease. These connections are summarized in Table 1 [56].

## 5. Other Inflammatory Conditions and CVD Progression

As previously discussed, chronic PD has been linked to accelerated atherosclerosis and elevated CVD risk. However, it is also worth mentioning that there are other systemic inflammatory conditions like rheumatoid arthritis (RA), lupus, and psoriasis that can drive up systemic cytokines like IL-6 and TNF-α (including acute phase reactants like CRP), which, in turn, promotes, endothelial dysfunction and atherogenesis and plaque formation [48,57]. More importantly, the magnitude of CVD correlates with the degree of SI. For example, lupus and untreated RA are highly inflammatory states, which can increase the risk of CVD by 2–3 times. This exceeds the inherent risk of DM, which can increase CVD risk by 24% (RR = 1.24). Other inflammatory conditions like psoriasis and, in this case, PD generally have lower SI severity. However, these conditions can still increase the risk of CVD by as much as PD has been linked to an increased risk of CVD, primarily from shared inflammatory pathways, at 50% [58,59]. These measurable findings emphasize that SI is the key contributor to CVD risk, which is why there is an increase in the utilization of anti-inflammatory therapies (e.g., IL-1 or IL-6 inhibitors or colchicine) [60,61].

## 6. Inflammation and CVD Risk: Evidence from Meta-Analyses and Cohort Studies

As already discussed, there is extensive evidence linking SI with increased CVD risk. In addition, a 2023 meta-analysis of 33 prospective studies confirmed that individuals with elevated inflammatory biomarkers (IL-6 or CRP levels) have a higher risk of developing CV events. Incremental increases in IL-6 or high-sensitivity CRP (hs-CRP) levels were associated with a 20% higher chance of developing CV-related events [62]. Observational studies showed that chronic PD can increase CVD risk by a pooled average of 34%, while long-term cohort studies (10+ years) show that CVD risk can increase as much as 200%. These findings consistently demonstrate that greater SI burden and chronicity can translate to significantly higher CVD risk (in the form of stroke, both cardioembolic and thrombotic types) [63].

## 7. The Impact of Cardiorespiratory Fitness on Periodontitis and CVD

Increasing physical activity (PA) and cardiorespiratory fitness (CRF) is known to decrease CVD risk and mortality [64,65,66]. Consequently, regular PA, which is formally known as exercise training (ET), has also been associated with a reduced prevalence of PD. For instance, a 2023 cross-sectional study involving 120 university students aged 18–29 in Shanghai examined the relationship between PA levels and OH. Using the International Physical Activity Questionnaire (IPAQ) to assess activity levels and the Community Periodontal Index (CPI) for OH status, that study found that participants meeting the recommended PA levels [≥600 metabolic equivalent (MET)] had a significantly lower prevalence of PD (60.7%) compared to those not meeting the said recommendations (91.7%) (*p* = 0.001). In addition, high-intensity PA was specifically associated with a lower incidence of PD (*p* = 0.006), while insufficient PA was significantly correlated with a higher likelihood of PD—yielding an OR = 5.293 (95% CI: 1.334–20.993; *p* = 0.018) after adjusting for factors such as gender, age, toothbrushing frequency, and flossing habits [67,68]. Supporting this, Ferreira et al. (2019) found that physically active individuals had a 33% lower likelihood of developing PD compared to inactive individuals (OR = 0.67; 95% CI: 0.56–0.81; *p* < 0.0001), further underscoring the protective effect of PA against periodontal inflammation [60].

On the contrary, a study published in the Journal of Clinical Periodontology examined 72 healthy, non-smoking, sedentary men aged 45 to 65 years and found that those with moderate to severe periodontitis (PD) had significantly lower peak oxygen uptake (VO_2_max) compared to those with no or mild PD (*p* = 0.026). These findings suggest a potential association between more severe periodontal inflammation and reduced cardiorespiratory fitness (CRF). However, given the study’s cross-sectional design, causality cannot be established, meaning that it remains unclear whether PD contributes to reduced CRF or if lower CRF predisposes individuals to more severe PD [61]. Building on these findings, a 2021 study further explored the relationship between PD severity and CRF among non-smoking men aged 45 to 74 years. Using data from two independent cross-sectional population-based studies (SHIP-1 and SHIP-Trend-0), the researchers found that individuals with moderate to severe PD—measured by mean pocket probing depth—consistently exhibited lower VO_2_max and shorter exercise duration compared to those with less severe PD. These results reinforce the notion that greater periodontal disease severity may be independently associated with diminished exercise capacity, suggesting that poor oral health could negatively influence overall fitness levels [62].

Regular PA has shown certain anti-inflammatory effects that benefit CV health. Studies indicate that individuals engaging in higher volumes of ET experience significantly lower SI. For instance, endurance athletes with more intensive training regimens show markedly reduced CRP concentrations (r = −0.32, *p* = 0.002) [69]. A 2007 study by Kuo et al. [70] analyzed NHANES 1999–2002 data (*n* = 1438 adults aged 20–49 years) and stratified participants into quartiles based on CRP concentrations. The findings showed that individuals in the highest CRP quartile (>0.24 mg/dL) had 3.42 times higher odds of having lower cardiorespiratory fitness (CRF) compared to those in the lowest CRP quartile (<0.06 mg/dL) (95% CI: 1.52–7.68, *p* =0.002) [67]. Adults who engage in routine exercise, including resistance ET (rET) multiple times per week and regular aerobic ET (aET), have approximately 40–50% lower odds of elevated CRP levels compared to inactive individuals (adjusted OR ≈ 0.5, *p* < 0.01) [68]. By reducing chronic inflammation and enhancing endothelial function, physical activity plays a crucial role in mitigating CVD risk. This underscores a key connection: an active lifestyle lowers SI, ultimately contributing to improved CV health. In addition to its anti-inflammatory effects, ET plays a crucial role in shaping lipid profiles. A longitudinal study by Park et al. (2015) [71] (*n* = 11,418 men) examined lipid trajectories over several decades and found that total cholesterol (TC), triglycerides (TG), low-density lipoprotein cholesterol (LDL-C), and non-high-density lipoprotein cholesterol (non-HDL-C) followed an inverted U-shaped trajectory with aging, increasing until the mid-40s to early 50s before gradually declining. Notably, individuals with lower CRF developed DLD significantly earlier—up to 15 years earlier—than those with higher CRF. Additionally, men with higher CRF had a lower risk of elevated TC levels compared to their low-CRF counterparts, highlighting the protective role of physical fitness against adverse lipid changes over time (Figure 1).

The study found that men with lower CRF developed DLD up to 15 years earlier, particularly among younger and middle-aged groups, highlighting the protective effect of high CRF [71]. Supporting this, a massive meta-analysis by Lin et al., which included 160 randomized controlled trials (RCTs) encompassing 7487 participants, demonstrated that ET significantly improved lipid profiles, increasing HDL-C (WMD = 2.32 mg/dL; 95% CI: 1.16–3.87; I^2^ = 87.5%; *p* < 0.001) and reducing triglycerides (WMD = –5.76 mg/dL; 95% CI: –10.63 to –0.89; I^2^ = 71.8%; *p* < 0.001). WMD represents the average treatment effect, while I^2^ indicates heterogeneity between studies. High I^2^ values suggest variation in populations, protocols, or measurements, which limits generalizability but does not undermine the overall finding—since the effect direction and significance were consistent—supporting ET as a robust intervention for improving lipid metabolism and reducing cardiometabolic risk [72].

In 2019, Lavie et al. observed a similar trend in their formal cardiac rehabilitation endurance training (CRET) program, which resulted in a 6% increase in HDL-C, a 15% reduction in triglycerides (TG), and a 2% decrease in LDL-C. Importantly, higher levels of CRF were strongly associated with better lipid profiles, and maintaining or improving CRF over time was linked to a lower incidence of DLD and improved survival outcomes. These findings highlight the critical role of both aET and rET in enhancing CRF, thereby reducing the risk of CVD and metabolic disorders [73]. Breneman et al. adjusted for covariates (e.g., sex, baseline BMI, baseline METs, smoking status, HTN status, etc.) and analyzed data from the Aerobics Center Longitudinal Study (ACLS) (*n* = 9651, aged ≥20 years). Their findings demonstrated that individuals with high baseline fitness had 43% lower odds of developing atherogenic DLD compared to those with low baseline fitness (OR = 0.57, 95% CI: 0.37 to 0.89, *p* < 0.001). However, after further adjustment for baseline lipid levels (HDL-C, LDL-C, TG), the association became statistically nonsignificant (OR = 1.03, 95% CI: 0.65–1.64, *p* < 0.001), confirming that fitness alone may not be an independent predictor of DLD when lipid profiles are accounted for. Still, these findings support the role of fitness in indirectly mitigating DLD through favorable lipid regulation over time [74].

CRF likewise attenuates CVD risk, portraying an intricate relationship between CRF, SI, and CVD outcomes. In conjunction, CRF has been demonstrated to impact CV outcomes beyond lipid level regulation. Hung et al., in the Henry Ford Exercise Testing Project, evaluated 33,204 patients with hyperlipidemia who underwent physician-referred treadmill stress testing. Their findings indicated that incremental increases of MET by 1 in CRF were associated with a 14% to 17% reduction in mortality risk, a 7% to 13% reduction in MI risk, and a 9% to 13% reduction in revascularization risk. These associations were independent of gender and statin therapy status [75]. Jiménez-Pavón et al. attribute a 1-MET increment in CRF to a 14–22% reduction in the incidence of sudden cardiac death (SCD). Specifically, each 1-MET increase was associated with a 13% lower risk of SCD (HR = 0.87; 95% CI: 0.83–0.90; *p* < 0.001). Overall, improvements in fitness levels are associated with a significant reduction in CVD and all-cause mortality, which should compel everyone to engage in routine ET [70,76].

## 8. Improving Cardiorespiratory Fitness

Mendoza et al. (2022) emphasize that structured ET programs—particularly those combining aET and rET—are not only highly effective in enhancing CRF but also leading to meaningful improvements in lipid biomarkers, reductions in SI, and overall cardiometabolic health [77]. The 2018 Physical Activity Guidelines (PAG) recommends that adults engage in at least 150 min per week of moderate-intensity aET or at least 75 min per week of vigorous-intensity aET [78]. However, the 2020 European Society of Cardiology recommends a more stringent duration of 300 min/wk of moderate-intensity aET. Nonetheless, shorter durations should be recommended for those who are severely deconditioned [79]. Due to concerns regarding the potential risks of adverse CV events linked to high-volume and high-intensity exercise therapy, gradual progression of aET duration is recommended since individuals who often exhibit the greatest need for aerobic conditioning are the ones who have the most reduced aerobic fitness, posing potential risks of adverse cardiac events linked to high-volume and high-intensity ET [80]. It is important to emphasize that the aforesaid risk is typically seen in individuals with low CRF but who engage in high volume and intensity ET beyond the recommended guidelines, often without adequate rest or recovery [80,81]. Concordantly, the second mode of ET is rET, and it remains a viable option for deconditioned individuals. Recommendations for non-aerobic or resistance exercise training (rET) vary, but current guidelines suggest attending two to three sessions per week. Each session should target multiple muscle groups, incorporating two to three sets of 8 to 12 repetitions using isometric contractions or lifts performed at moderate to vigorous intensity [78,79,82]. In 2022, Mendoza and Lavie discussed that both aET and rET independently enhanced CRF and muscular strength, respectively, and were associated with reduced mortality rates. However, they highlight that engaging in both modes of ET can lead to more pronounced improvements in DLD and CVD-related mortality compared to either ET modality alone; and Table 2 summarizes the aforementioned PAG guidelines [83].

To further underscore the importance of improving CRF, Kokkinos et al. published a large cohort study (*n* = 110,637), which demonstrated an inverse association between CRF and all-cause mortality across age, sex, and race. Individuals with the lowest fitness levels and traditional CVD comorbidities had a four-fold higher risk of all-cause mortality compared to those with the highest CRF levels HR = 4.09; 95% CI: 3.94–4.24; *p* < 0.001). Moreover, for every 1-MET increase in exercise capacity, the adjusted HR for mortality decreased to 0.86 (95% CI: 0.85–0.87; *p* < 0.001) across the entire cohort. These findings reinforce that improving CRF significantly reduces all-cause mortality risk and contributes meaningfully to CVD prevention [84].

While RCTs or longitudinal cohort studies directly linking PD and reduced CRF are limited, cross-sectional evidence suggests a strong association. Variation in how PD and CRF are measured makes cross-study comparison difficult, and the observational nature of most available data limits causal inference. However, a multivariate analysis from the CHIEF cohort by Tsai et al. found that poor OH nearly doubled the risk of poor physical performance in physically active military men, OR = 1.94 (95% CI: not reported; *p* = 0.04), suggesting broader systemic implications. As the connection between OH and CV outcomes becomes clearer, it is plausible that oral health may also influence CRF. Integrating dental assessments and oral hygiene promotion into preventive care strategies may enhance CRF, reduce SI, and lower CV risk. Further standardized, longitudinal studies are warranted to confirm these relationships [26].

## 9. Future Directions and Clinical Applications

### 9.1. Anti-Inflammatory Therapies and Lifestyle Interventions for Periodontal and Cardiovascular Disease

As SI is a key driver of both PD and CVD, there is growing interest in targeting inflammatory pathways to mitigate disease progression. Emerging therapies, such as IL-6 inhibitors, colchicine, and PCSK9 inhibitors, have demonstrated promise in reducing cardiovascular events by modulating inflammatory cascades. Future studies should investigate whether these agents also confer benefits in managing PD-related inflammation. In parallel, lifestyle interventions—including structured exercise programs and dietary modifications—have demonstrated the potential to reduce SI and improve both periodontal and CV outcomes. ET exerts well-documented anti-inflammatory effects by lowering CRP and IL-6 levels while also enhancing endothelial function. Promoting PA as a dual-purpose intervention for PD and CVD offers a promising non-pharmacologic strategy to improve overall health.

### 9.2. Interdisciplinary Approaches: Advocate for Collaboration Between Dental, Medical, and Fitness Professionals

Given the substantial overlap in risk factors for PD and CVD—particularly SI, metabolic dysfunction, and lifestyle factors—a multidisciplinary care model is warranted. Cardiologists, dentists, primary care physicians, and exercise physiologists should collaborate to implement comprehensive screening protocols. For instance, periodontal health assessments could become a routine component of cardiovascular risk stratification, particularly for individuals with established CVD, DM, or other pro-inflammatory conditions. Similarly, dentists could play a proactive role in identifying patients at high cardiovascular risk by incorporating non-invasive cardiovascular assessments, such as blood pressure screening and inflammatory biomarker testing, into routine dental visits. Integrating oral health into preventive cardiology programs may provide a cost-effective strategy to reduce the burden of CVD.

### 9.3. Public Health Implications: Stress the Importance of Awareness Campaigns for Better Oral and Cardiovascular Health

Despite the clear association between PD and SI, public awareness of the oral–systemic health connection remains low. Educational campaigns targeting both healthcare professionals and the general public are critical to improving the recognition of PD as a modifiable risk factor for CVD. Initiatives should emphasize the importance of regular dental check-ups, optimal oral hygiene practices, smoking cessation, and lifestyle modifications as part of a comprehensive cardiovascular prevention strategy. At a policy level, integrating dental and medical healthcare systems could improve access to periodontal care for populations at high risk of CVD. Expanding insurance coverage to include periodontal evaluations and treatments for individuals with cardiovascular risk factors may yield significant public health benefits.

## 10. Conclusions

What if a healthier smile could mean a stronger heart? The emerging science connecting PD and CVD is reshaping how we understand inflammation and its systemic impact. PD, once considered a localized dental issue, is now a proven contributor to SI—thereby driving atherogenesis, endothelial dysfunction, and subsequent major adverse cardiovascular events. Elevated inflammatory markers like CRP and IL-6 highlight the inflammatory bridge between poor OH and heart disease. Meta-analyses show that PD raises the risk of MI, stroke, and all-cause mortality by as much as 31%. On the other hand, improved OH, structured ET, and better CRF not only reduce PD severity but also lower CVD risk. The combination of aET and rET has been shown to produce superior outcomes compared to either alone, delaying DLD, improving lipid profiles, and extending life expectancy. Inflammation-targeted therapies and interventions, including IL-6 inhibitors and colchicine, are gaining attention not only in cardiology but, potentially, in dentistry as well.

The intricate relationship between PD and SI calls for a paradigm shift in both clinical practice and public health strategy. Future research must focus on establishing causality, identifying novel therapeutic targets, and implementing interdisciplinary models that bridge dentistry, cardiology, and preventive medicine. By recognizing PD as both a marker and a contributor to CV risk, clinicians can adopt a more holistic and proactive approach. Integrating OH screenings into CV risk stratification and incorporating non-invasive CV assessments in dental visits can create powerful touchpoints for early intervention. Public health campaigns must raise awareness, especially in at-risk populations. With coordinated action, we can move toward a future where oral health is treated as a cornerstone of chronic disease prevention. In doing so, we not only reduce the burden of CVD but also empower patients to take charge of their well-being, starting with one brush… and one rep at a time.

## Figures and Tables

**Figure 1 biomedicines-13-01512-f001:**
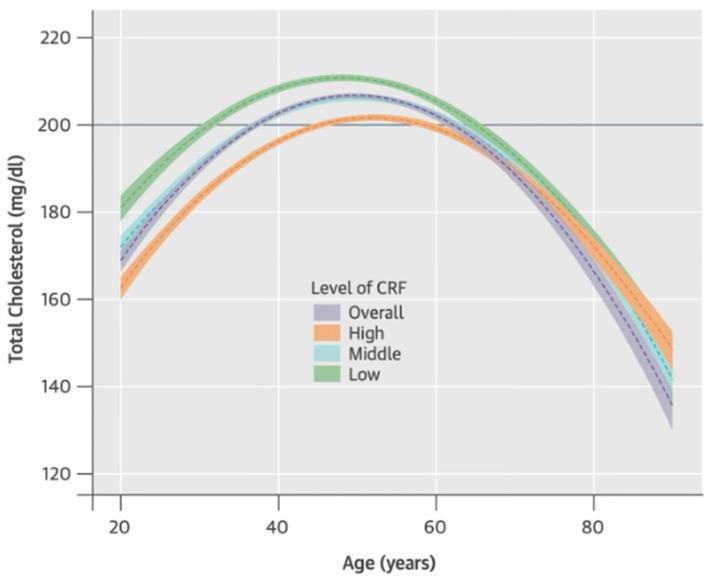
The purple dashed line represents the crude overall trajectory; the other dashed lines represent each category of cardiorespiratory fitness (CRF) (orange for high CRF, light blue for middle CRF, and green for low CRF) with shaded areas representing 95% confidence intervals for each CRF category. The solid blue horizontal line indicates the established abnormal cutoff of 200 mg/dl for total cholesterol (taken from Park et al., 2015 [71]).

**Table 1 biomedicines-13-01512-t001:** Key points connecting PD and CVD (taken from Priyamvara et al., 2020) [56].

Bacterial Translocation in PD	Inflammation in PD and CVD
In periodontal disease, periodontal vasculature is dilated and facilitates bacteremia	Periodontitis produces elevated levels of inflammatory mediators (CRP, haptoglobin, fibrinogen, TNF-α, IL-6, fibrinogen)
When the sulcular epithelium is disturbed, bacteria translocate into the bloodstream	Inflammatory mediators may modify serum lipid, inflammatory cell, and endothelial cell receptors, contributing to plaque development
Beginning with PGE2, an inflammatory cascade creates a reservoir of gram-negative bacteria and pro-inflammatory products (LPS) in the periodontium	Elevated CRP correlates with an increase in CIMT in PD patients, ultimately conferring increased stroke risk
Common dental plaque anaerobic bacteria have been found particularly in locations where atherosclerotic plaques are found (coronary arteries)	Patient with periodontitis has higher levels of LPS-LDL and oxLDL, which are modified lipid forms that promote atheroma development
Pathogens induce cross-reactive antibodies, which then produce inflammation systemically and within atheromas	Salivary assays for 8-hydroxyguanosine, malondialdehyde, protein carbonyl assay, and total antioxidant capacity are elevated in both chronic PD and ACS patients

**Table 2 biomedicines-13-01512-t002:** Summarized exercise prescription for adults based on the 2018 Physical Activity Guidelines (PAG) for Americans, 2nd edition, and the 2016 ESC Exercise Guideline. Adapted from [83].

Category	Exercise Type	Description
Mode	Aerobic ET	Brisk walking, running/jogging, swimming, bicycling, stair climbing, rowing, aerobic dancing, skiing
	Resistance ET	Lifting weights, resistance/elastic bands, bodyweight exercises, heavy gardening, calisthenics
Duration	Aerobic ET	150–300 min/week (moderate intensity); 75–150 min/week (vigorous intensity); or an equivalent combination of moderate to vigorous intensity
	Resistance ET	8 to 12 repetitions to fatigue; at least 1 set for all muscle groups; 60 to 80% of single maximum repetition (70% in elderly)
Frequency	Aerobic ET	Most days of the week (preferably 6–7 days/week)
	Resistance ET	2–3 non-consecutive days/week
Intensity	Aerobic ET	Moderate to vigorous intensity
	Resistance ET	Moderate to vigorous intensity

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
