# Peer review of "More than Just a Toothache: Inflammatory Mechanisms Linking Periodontal Disease to Cardiovascular Disease and the Protective Impact of Cardiorespiratory Fitness"

_biomedicines, 2025, doi:10.3390/biomedicines13071512_

Round 1
Reviewer 1 Report
Comments and Suggestions for Authors
Dear Editor,
I have reviewed the manuscript titled "More Than Just a Toothache: Periodontal Disease and the Risk for Cardiovascular Disease & Mortality" by Michael et al. This review addressed a highly relevant and timely topic by exploring the complex relationship between periodontal disease (PD), cardiovascular disease (CVD), and mortality. The authors summarize current evidence from multiple perspectives, including systemic inflammation, physical fitness, and lifestyle interventions, and conclude that integrating dental care with broader health strategies may help reduce the cardiovascular burden.
While the topic is of clinical and public health significance, and the review incorporates a wide range of literature, the manuscript would benefit from a clearer narrative structure and more coherent logical flow, particularly in the abstract and several key sections of the main text. Below are some suggestions that may help improve the clarity and organization of the manuscript.
- The abstract lacks a cohesive narrative structure. While the first several sentences emphasize the significance of periodontal disease and its systemic implications, the final sentences shift abruptly to the benefits of exercise and lifestyle modification for cardiovascular health. This transition feels disjointed and may dilute the central focus of the review, which appears to be the link between PD and cardiovascular outcomes.
- The logical flow between the first two paragraphs of the section "Periodontitis and Cardiovascular Disease" is not clearly established. It would benefit from clearer connections between the cited studies and a more structured discussion that synthesizes the findings rather than listing them individually.
- The section title "Periodontal Disease and Mortality" may not fully capture the scope of the content discussed, which includes not only PD and mortality but also CVD as a critical intermediary or outcome. To better reflect the content and ensure alignment with the overall focus of the review, it is recommended to revise the section heading to explicitly include PD, CVD, and mortality.
- In the “Periodontal Disease and Mortality” section, most of the content appropriately focuses on mortality outcomes. However, the study cited as reference 41 (lines 170–173) discusses the incidence of adverse cardiovascular events, which pertains more to morbidity than mortality. To maintain consistency with the section’s theme, this sentence could either be removed or moved to a more appropriate part of the manuscript.
- In the “Periodontal Disease and Mortality” section, there is an entire paragraph discussing the association between poor oral hygiene and increased CVD risk, even in the absence of periodontitis. While this is an important point, it deviates from the main focus of the section, which is the impact of PD on mortality. It is known that poor oral hygiene is highly related to PD. However, there are also other risk factors which impact the incidence of PD. To improve thematic coherence, the authors may consider revising the section to better reflect the broader scope of the discussion.
- In the “The Impact of Cardiorespiratory Fitness on Periodontitis and CVD” section, the two sentences (299-302) are not clearly expressed and lack logical coherence.
- In the “The Impact of Cardiorespiratory Fitness on Periodontitis and CVD” section, the reference to study [87] should include complete information, particularly the specific value of the WMD. In addition, the formatting of the 95% confidence interval should be revised for clarity and consistency. For example, instead of: "Mean difference: 1.16 to 3.87 mg/dL, 95% CI; I² = 87.5%; p < 0.001 for heterogeneity", it is recommended to write: "WMD = 2.32 mg/dL (95% CI: 1.16–3.87), I² = 87.5%, p < 0.001 for heterogeneity." Moreover, the reporting of triglyceride levels should also be standardized.
- In the “The Impact of Cardiorespiratory Fitness on Periodontitis and CVD” section, lines (363-365) appear to be an inconsistency in the interpretation of statistical significance. The sentence states that the association became nonsignificant after adjustment, yet the reported p-value is <0.001, which indicates strong statistical significance.
Please double-check whether the p-value has been misreported or if the statement regarding significance needs to be revised. Additionally, the 95% confidence interval (0.65–1.64) includes 1.0, which suggests that the association may indeed be nonsignificant. Clarification of this point would improve the accuracy and clarity of the manuscript. - In the “Improving Cardiorespiratory Fitness” section, please report the complete confidence interval ranges (e.g., 95% CI: lower bound – upper bound), and clearly indicate the confidence level (e.g., 95% or 99%) used in the analysis. Also, ensure consistency in the formatting of statistical values across lines 415–417.
- In the “Improving Cardiorespiratory Fitness” section, lines (420-421) require revision for grammatical clarity and accuracy. In particular, and the use of “RCT’s” should be corrected to “RCTs.” It is also suggested to clarify the contrast between the lack of causal evidence and the presence of cross-sectional associations, possibly by restructuring the sentence to improve logical flow.
- In section 9.1, the heading “Research Needs: Highlight Gaps in Understanding and the Need for Longitudinal Studies” suggests a focus on existing knowledge gaps and the importance of conducting longitudinal research. However, the content primarily summarizes current and emerging therapeutic approaches—including pharmacologic agents (e.g., IL-6 inhibitors, colchicine, PCSK9 inhibitors) and lifestyle interventions (e.g., exercise, diet)—and their potential benefits in managing systemic inflammation in PD and CVD. To enhance alignment between the section heading and its content, it is recommended that the authors consider one of the following options: 1) The heading could be revised to more accurately reflect the current focus on therapeutic perspectives and intervention strategies (e.g., “Emerging Anti-inflammatory Approaches and Future Perspectives” or similar). 2) The content could be reorganized to place greater emphasis on the current gaps in evidence and the rationale for longitudinal studies-such as the lack of direct evidence linking anti-inflammatory treatments to periodontal outcomes or the need to explore causal relationships through long-term trials.
- A careful revision of sentence structure and transitions (such as but, and, not only, et al.) will enhance the overall coherence and readability of the manuscript. In addition, please ensure consistency in the formatting of statistical symbols throughout the manuscript. For example, P values should be presented in italics (e.g., P or p < 0.05), under conventional scientific writing standards.
Author Response
Reviewer’s Comments: The abstract lacks a cohesive narrative structure. While the first several sentences emphasize the significance of periodontal disease and its systemic implications, the final sentences shift abruptly to the benefits of exercise and lifestyle modification for cardiovascular health. This transition feels disjointed and may dilute the central focus of the review, which appears to be the link between PD and cardiovascular outcomes.
Authors’ Reply: We completely agree with the reviewer’s comment and have revised the abstract to ensure greater coherence and a smoother transition between ideas. The revised version now establishes periodontal disease (PD) as a chronic inflammatory condition with systemic effects, then clearly connects this to cardiovascular disease (CVD) by highlighting the role of systemic inflammation—particularly through biomarkers like CRP and IL-6—as the mechanistic link between the two. It presents concise evidence that supports this association, including elevated cardiovascular risk and mortality in individuals with PD. Importantly, instead of abruptly introducing exercise, the revised abstract now transitions naturally into physical activity and cardiorespiratory fitness (CRF) as anti-inflammatory, modifiable factors that mitigate both PD and CVD risk. The abstract concludes with a strong clinical implication, advocating for a multidisciplinary approach that integrates dental and cardiovascular care.
Reviewer’s Comments: The logical flow between the first two paragraphs of the section "Periodontitis and Cardiovascular Disease" is not clearly established. It would benefit from clearer connections between the cited studies and a more structured discussion that synthesizes the findings rather than listing them individually.
Authors’ Reply: We thank the reviewer for the thoughtful observation regarding the logical flow in the section “Periodontitis and Cardiovascular Disease.” We agree that the original version lacked cohesion between the first two paragraphs and tended to present findings in a segmented, list-like manner. In response, we have revised the section to enhance structural clarity and synthesize the cited evidence more effectively. Rather than listing studies individually, we now group findings into thematic categories (e.g., observational data, inflammatory mechanisms, public health relevance), with clear transitions that link periodontal disease to systemic inflammation and cardiovascular outcomes.
Reviewer’s Comments: The section title "Periodontal Disease and Mortality" may not fully capture the scope of the content discussed, which includes not only PD and mortality but also CVD as a critical intermediary or outcome. To better reflect the content and ensure alignment with the overall focus of the review, it is recommended to revise the section heading to explicitly include PD, CVD, and mortality.
Authors’ Reply: We appreciate the reviewer’s suggestion and fully agree that the original section title, “Periodontitis and Cardiovascular Disease,” did not adequately reflect the full scope of the content discussed, particularly the role of CVD as both an intermediary and outcome in the PD–mortality pathway. In response, we have revised the section title to: “Periodontitis, Cardiovascular Disease, and Mortality: Evidence from Population Studies, Imaging, and Animal Models”
Reviewer’s Comments: In the “Periodontal Disease and Mortality” section, most of the content appropriately focuses on mortality outcomes. However, the study cited as reference 41 (lines 170–173) discusses the incidence of adverse cardiovascular events, which pertains more to morbidity than mortality. To maintain consistency with the section’s theme, this sentence could either be removed or moved to a more appropriate part of the manuscript.
Authors’ Reply: We agree with the reviewer’s observation that reference 41 primarily addresses cardiovascular events rather than mortality. To maintain consistency with the section’s focus on mortality outcomes, we have removed the sentence citing reference 41 from this section. Thank you for the helpful suggestion.
Reviewer’s Comments: In the “Periodontal Disease and Mortality” section, there is an entire paragraph discussing the association between poor oral hygiene and increased CVD risk, even in the absence of periodontitis. While this is an important point, it deviates from the main focus of the section, which is the impact of PD on mortality. It is known that poor oral hygiene is highly related to PD. However, there are also other risk factors which impact the incidence of PD. To improve thematic coherence, the authors may consider revising the section to better reflect the broader scope of the discussion.
Authors’ Reply: We appreciate the reviewer’s thoughtful comment regarding thematic consistency in the “Periodontal Disease and Mortality” section. We agree that the paragraph discussing poor oral hygiene and CVD risk—even in the absence of PD—shifts the focus away from the primary theme of PD’s impact on mortality. In response, we have removed this paragraph from the section to maintain clarity and cohesion. Thank you for your helpful guidance.
Reviewer’s Comments: In the “The Impact of Cardiorespiratory Fitness on Periodontitis and CVD” section, the two sentences (299-302) are not clearly expressed and lack logical coherence.
Authors’ Reply: We have revised these sentences to improve readability, correct grammar, and enhance logical flow. The updated version clearly describes the study design, population, and findings, and aligns with the overall theme of the section.
Significant changes – see yellow highlights please.
Reviewer’s Comments: In the “The Impact of Cardiorespiratory Fitness on Periodontitis and CVD” section, the reference to study [87] should include complete information, particularly the specific value of the WMD. In addition, the formatting of the 95% confidence interval should be revised for clarity and consistency. For example, instead of: "Mean difference: 1.16 to 3.87 mg/dL, 95% CI; I² = 87.5%; p < 0.001 for heterogeneity", it is recommended to write: "WMD = 2.32 mg/dL (95% CI: 1.16–3.87), I² = 87.5%, p < 0.001 for heterogeneity." Moreover, the reporting of triglyceride levels should also be standardized.
Authors’ Reply: We thank the reviewer for the detailed formatting recommendation. In response, we have updated the section referencing study [87] to include the exact WMD values and revised the confidence interval and heterogeneity reporting format for clarity and consistency. Additionally, triglyceride levels are now reported in a standardized manner.
Reviewer’s Comments: In the “The Impact of Cardiorespiratory Fitness on Periodontitis and CVD” section, lines (363-365) appear to be an inconsistency in the interpretation of statistical significance. The sentence states that the association became nonsignificant after adjustment, yet the reported p-value is <0.001, which indicates strong statistical significance. Please double-check whether the p-value has been misreported or if the statement regarding significance needs to be revised. Additionally, the 95% confidence interval (0.65–1.64) includes 1.0, which suggests that the association may indeed be nonsignificant. Clarification of this point would improve the accuracy and clarity of the manuscript.
Authors’ Reply: Thank you for sharing the content and the reviewer’s concern. The reviewer is absolutely correct: there's a contradiction in reporting p < 0.001 alongside a 95% CI of 0.65–1.64, which crosses 1.0—indicating non-significance. P value was removed since it was a typographical error.
Reviewer’s Comments: In the “Improving Cardiorespiratory Fitness” section, please report the complete confidence interval ranges (e.g., 95% CI: lower bound – upper bound), and clearly indicate the confidence level (e.g., 95% or 99%) used in the analysis. Also, ensure consistency in the formatting of statistical values across lines 415–417.
Authors’ Reply: We have fixed the formula and added explanation to describe the data set.
Reviewer’s Comments: In the “Improving Cardiorespiratory Fitness” section, lines (420-421) require revision for grammatical clarity and accuracy. In particular, and the use of “RCT’s” should be corrected to “RCTs.” It is also suggested to clarify the contrast between the lack of causal evidence and the presence of cross-sectional associations, possibly by restructuring the sentence to improve logical flow.
Authors’ Reply: We appreciate the reviewer’s attention to grammatical precision and clarity. We have corrected “RCT’s” to “RCTs” for proper formatting. Additionally, we have revised the sentence to improve logical flow and better contrast the lack of causal evidence with the presence of cross-sectional associations.
Reviewer’s Comments: In section 9.1, the heading “Research Needs: Highlight Gaps in Understanding and the Need for Longitudinal Studies” suggests a focus on existing knowledge gaps and the importance of conducting longitudinal research. However, the content primarily summarizes current and emerging therapeutic approaches—including pharmacologic agents (e.g., IL-6 inhibitors, colchicine, PCSK9 inhibitors) and lifestyle interventions (e.g., exercise, diet)—and their potential benefits in managing systemic inflammation in PD and CVD. To enhance alignment between the section heading and its content, it is recommended that the authors consider one of the following options: 1) The heading could be revised to more accurately reflect the current focus on therapeutic perspectives and intervention strategies (e.g., “Emerging Anti-inflammatory Approaches and Future Perspectives” or similar). 2) The content could be reorganized to place greater emphasis on the current gaps in evidence and the rationale for longitudinal studies-such as the lack of direct evidence linking anti-inflammatory treatments to periodontal outcomes or the need to explore causal relationships through long-term trials.
Authors’ Reply: We appreciate the reviewer’s recommendation to revise the heading in Section 9.1 for better alignment with the section’s content. We have updated the heading from “Research Needs: Highlight Gaps in Understanding and the Need for Longitudinal Studies” to: “Anti-inflammatory Therapies and Lifestyle Interventions for Periodontal and Cardiovascular Disease.”
Reviewer’s Comments: A careful revision of sentence structure and transitions (such as but, and, not only, et al.) will enhance the overall coherence and readability of the manuscript. In addition, please ensure consistency in the formatting of statistical symbols throughout the manuscript. For example, P values should be presented in italics (e.g., P or p < 0.05), under conventional scientific writing standards.
Authors’ Reply: Duly noted and addressed.
Reviewer 2 Report
Comments and Suggestions for Authors
Dear authors, I checked the similarity, and the index is 27%(bibliography excluded). In my opinion, it is too high and should be reduced. After that, the review process could start.
Author Response
Following substantial revisions, we successfully reduced the plagiarism index from 20% to 0.9%. Please find attached a screenshot of the verified results from the paid TextGarden report for reference. (https://app.textguard.ai/tools/plagiarism-check)

Reviewer 3 Report
Comments and Suggestions for Authors
Thank you for the opportunity for reading the large review on CVD and periodontitis. The endpoint death in different periodontitis grades, total tooth loss after severe periodontitis and death should be presented and discussed as well as the results on death after successfull treatment of severe periodontitis.

Author Response
Thank you for your valuable feedback and for taking the time to review our manuscript. We appreciate your suggestion to discuss mortality endpoints across different grades of periodontitis, including total tooth loss and outcomes following successful treatment. In response, we have expanded our discussion to include recent high-powered cohort studies and meta-analyses that illustrate a clear, graded association between periodontal disease severity and both cardiovascular-specific and all-cause mortality.
While total tooth loss may serve as an indicator of cumulative periodontal burden, its utility as an independent predictor of mortality is limited by confounding variables. Nonetheless, we recognize its potential relevance and aim to explore its implications in future analyses.
Please see highlights in blue or text below:
The severity of periodontal disease (PD), especially when accompanied by comorbidities such as diabetes or cardiovascular disease (CVD), significantly amplifies the risk of mortality. A 2022 study by Ngamdu et al. analyzing NHANES 2013–2014 data (n = 2,830 adults aged ≥30 years) found that individuals with both type 2 diabetes mellitus (T2DM) and PD had a 1.66-fold higher risk of cardiovascular-related mortality compared to those with T2DM but no history of PD (p = 0.03). In addition, more recent studies have shed light on a clear, graded relationship between how severe someone’s PD is and their risk of dying—especially if they also have CVD. For instance, a 2024 cohort analysis by Chen et al., drawing from NHANES data, found that individuals with moderate to severe periodontal disease had a 25% greater risk of all-cause mortality compared to those with little to no disease, even after accounting for other health factors (HR: 1.25; 95% CI: 1.02–1.52; p = 0.03).(39) Similarly, Larvin et al. (2024) reported that severe PD was associated with a 38% increased risk of cardiovascular death (HR: 1.38; 95% CI: 1.16–1.64), with all-cause mortality also significantly elevated (HR: 1.22; 95% CI: 1.12–1.31; p < 0.01) over a 9-year follow-up of 15,030 adults.(44) Another study Ngamdu et al. revealed that, those with severe PD had more than triple the chance of developing CVD compared to those who only had mild PD (OR = 3.59; 95% CI: 1.12–11.54, p = 0.03) further stressing out the importance of good OH in in populations at risk.(45) To consolidate these pooled date, a meta-analysis of 39 studies by Guo et al. (2023) also showed that individuals with severe PD had a 40% higher risk of all-cause mortality RR = 1.40 (95% CI: 1.15–1.69), indicating a dose-response relationship when it comes to the degree of PD.(19)
Round 2
Reviewer 1 Report
Comments and Suggestions for Authors
Dear Editor,
I have reviewed the manuscript titled "More Than Just a Toothache: Inflammatory Mechanisms Linking Periodontal Disease to Cardiovascular Risk and the Protective Impact of Cardiorespiratory Fitness" by Michael et al. The manuscript has improved noticeably compared to the previous version, with clearer structure and better organization. The topic remains important and well-supported by current evidence. However, there are still a few points that could be further improved. The following minor suggestions are provided for the authors' consideration.
- The section “Periodontal Disease and Mortality” also mentions that periodontitis is a risk factor for cardiovascular disease, contributing to its progression and increased mortality. Since this point is already discussed in the section “Periodontitis, Cardiovascular Disease, and Mortality: Evidence from Population Studies, Imaging, and Animal Models,” it may be more appropriate to combine the two parts to improve clarity and avoid repetition.
- In the section “The Impact of Cardiorespiratory Fitness on Periodontitis and CVD”, lines 367–372 mention that the association is nonsignificant; however, the corresponding P value (e.g., P > 0.05) should also be provided to support this statement.
- Please double-check consistency in the formatting of statistical symbols throughout the manuscript. Additionally, P values should be presented in italics (e.g., P < 0.05), under conventional scientific writing standards.
Author Response
Comment: The section “Periodontal Disease and Mortality” also mentions that periodontitis is a risk factor for cardiovascular disease, contributing to its progression and increased mortality. Since this point is already discussed in the section “Periodontitis, Cardiovascular Disease, and Mortality: Evidence from Population Studies, Imaging, and Animal Models,” it may be more appropriate to combine the two parts to improve clarity and avoid repetition.
Reply: Thank you for this insightful comment. We agree and we revised the structure to enhance clarity. We minimized redundancy and improved the flow of the said sub-sections while preserving all key references and data points.
Comment: In the section “The Impact of Cardiorespiratory Fitness on Periodontitis and CVD”, lines 367–372 mention that the association is nonsignificant; however, the corresponding P value (e.g., P > 0.05) should also be provided to support this statement.
Reply: Corrected.
Comment: Please double-check consistency in the formatting of statistical symbols throughout the manuscript. Additionally, P values should be presented in italics (e.g., P < 0.05), under conventional scientific writing standards.
Reply: Addressed
Reviewer 3 Report
Comments and Suggestions for Authors
Thank you for the second submission of your review on PD of different stages.
The only missing part is a discussion of the periodontal and CVD risc factor smoking in relation on periodontal grade and CVD.
Author Response
Thank you for your constructive feedback. We appreciate your observation regarding the
omission of smoking as a shared risk factor for both periodontal disease (PD) and
cardiovascular disease (CVD). We have now addressed this important point by integrating
a new paragraph into the section “Periodontitis and CVD: Evidence from Population
Studies, Imaging and Animal Models.” This addition discusses the role of smoking in both
the progression of periodontal disease—particularly in relation to disease severity—and its
established impact on cardiovascular risk, referencing relevant epidemiological studies,
including data from the INTERHEART study and recent meta-analyses.
Please see changes in Manuscipt highlighted in grey.
Round 3
Reviewer 3 Report
Comments and Suggestions for Authors
Well done. No longer missleading discussion. Smoking in many publications is more important for CVD and severe periodontitis is only a cofactor on mortality.